# Divided Attention: Unsupervised Multiple-object Discovery and Segmentation with Interpretable Contextually Separated Slots

## Abstract

We introduce a lightweight method to segment the visual field into independently moving regions that requires no ground truth or manual supervision and can perform inference in real-time. The model consists of an adversarial conditional encoder-decoder architecture with interpretable latent variables, obtained by modifying the Slot Attention architecture to use the image as context to decode optical flow, but without attempting to reconstruct the image itself. One modality (flow) feeds the encoder to produce latent codes (slots) for each individual object, whereas the other modality (image) conditions the decoder to generate the first (flow) from the slots. This design frees the representation from having to encode complex nuisance variability in the image due to, for instance, illumination and reflectance properties of the scene. Since customary autoencoding based on minimizing the reconstruction error does not preclude the entire flow from being encoded into a single slot, we design the loss with an adversarial criterion. The resulting min-max optimization fosters the separation of objects and their assignment to different attention slots, leading to Divided Attention (DivA). DivA outperforms recent unsupervised multi-object motion segmentation methods while tripling run-time speed up to 104FPS and reducing the performance gap from supervised methods to 12% or less. DivA can handle different numbers of objects and different image resolutions at training and test time, is invariant to the permutation of object labels, and does not require explicit regularization. The code will be publicly available upon paper publication.

## 1 Introduction

The ability to segment the visual field by motion is so crucial to survival that we share it with our reptilian ancestors. A successful organism should not take too long to learn how to spot predators or obstacles and surely should require no supervision. Yet, in the benchmarks we employ to evaluate moving object discovery, the best-performing algorithms are either trained using large annotated datasets Lin et al. (2014) or simulation environments with ground truth Xie et al. (2022), or both Dave et al. (2019). Recently, alternative approaches Wang et al. (2022); Bielski & Favaro (2022); Seitzer et al. (2023); **?**); Ponimatkin et al. (2023) have been proposed leveraging powerful self-supervised image features Caron et al. (2021); Oquab et al. (2023) that do not rely on explicitly labeled data. Such approaches often exploit the bias of human-collected "object-centric" datasets where a single object is prominently featured, sometimes centered, and viewed from non-accidental vantage points. While one could argue that, since available, we might as well exploit such pre-trained models with human viewpoint selection bias, we are interested in exploring how far one can push the process of training *ab-ovo* without purposefully collected data and frame selection biases from an entity external to the model. Accordingly, we forgo the assumption that there is a "dominant" or "primary" object in the scene. Instead, we can have any number of independently moving agents.

We have thus far avoided a formal definition of "objects": Since we do not wish to rely on the definition implicit in the human selection of the shot, we define objects primitively as *regions of an image whose corresponding motion is unpredictable from their surroundings.* Objects thus can be interpreted as the projection of Gibson's "detached objects" Gibson (1978) onto the image, and formalized via a variational principle similar to Contextual Information Separation (CIS) Yang et al.

| Image | Flow | Recon. Flow | Slot 1 | Slot 2 | Slot 3 | Slot 4 | Segmentation |
|-------|------|-------------|--------|--------|--------|--------|--------------|

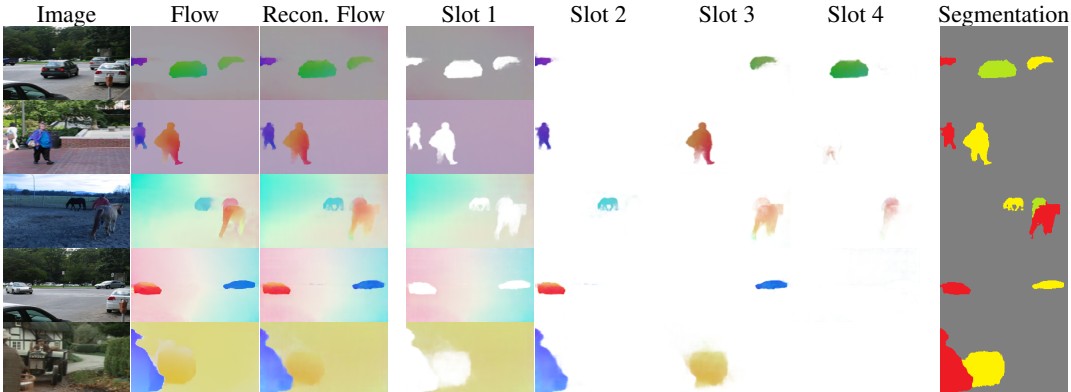

Figure 1: **Multi-region moving object discovery on real data.** *Our method discovers moving objects in videos without any pre-trained features, human annotation, or supervision. The slots are randomly initialized, so as to be permutable. We showcase them in this particular order for ease of visualization.*

(2019). However, CIS mostly applies to binary segmentation where a dominant "foreground object" is present. In a sense, our work can be thought of as combining CIS with Slot Attention to enable bootstrapped discovery of multiple moving objects. Such a combination requires modifying and generalizing both the Slot Attention architecture and the CIS training loss in non-trivial ways.

Specifically, we modify the Slot Attention Networks (SAN) architecture of Locatello et al. (2020), to use an image and its corresponding motion field as input. Ideally, each slot would be interpretable as an independently moving object. Current SANs minimize the reconstruction error, which does not impose an explicit bias to separate independent objects into slots. We hypothesize that the adversarial CIS loss can foster *"divided attention"* within the activations. However, naively combining CIS and SANs leads to disappointing results. Instead, we modify the architecture to incorporate *cross-modal generation*: We force the model to reconstruct the flow, *but not the image itself*, since most of its complexity can be ascribed to nuisance variability. We enable the image to *guide* the flow reconstruction. This is done by incorporating a Conditional Prior Network Yang & Soatto (2018), that models the distribution of flows compatible with the given image, and modifying the architecture to use a *conditional cross-modal decoder*. The resulting method *Divided Attention* (DivA) consists of a multi-modal architecture where one modality (flow) goes into the main encoder that feeds the slots, and the other (image) goes into the decoder that generates the flow (Fig. 2), trained adversarially with a loss (2) that captures both cross-modal generation and information separation. To the best of our knowledge, DivA is the first unsupervised moving object segmentation model that:

- handles a variable number of objects that can be changed at inference time without needing to re-train, and is invariant to permutations of object labels;
- uses the mean-squared error (MSE) as the base loss and does not need explicit regularization;
- requires only single images and corresponding flows as input, and operates in real-time;
- does not require any pre-trained image features from external datasets.

The ability to bootstrap the discovery of objects without manually annotated, simulated, or strongly biased datasets, and to enable inference with minimal latency is consistent with biological evolutionary development. An added benefit is that DivA produces interpretable results, where the adversarial loss fosters coalescence of pixels and separation of objects in different slots. Those could be used to prime a static object detector, by providing strong object proposals to then group into classes.

DivA is tested on both multiple moving object datasets and binary segmentation datasets. DivA improves performance on DAVIS and SegTrack by 5% and 7% respectively, compared to recent unsupervised single-frame baselines, and closes the gap to batch-processing and supervised methods to 9.5% and 12% respectively, all while significantly improving inference speed: An embodiment of DiVA that matches the current state of the art improves speed by 200% (21FPS to 64FPS), and our fastest embodiment reaches 104FPS. Compared to the paragon method that processes the entire video frame to produce segments, often relying on pre-trained features, our model does not perform as well, especially on benchmarks that focus on "object-centric" scenes with strong selection biases. Nonetheless, the gap enables us to perform inference with low latency, for we do not need to process an entire batch to produce an outcome, and low computational cost enables real-time operation.

## 1.1 RELATED WORK

**Unsupervised moving object segmentation.**   Also referred to as "motion segmentation", early methods segment dense optical flow [1] Horn & Schunck (1981) by optimizing a designed objective through the evolution of a partial differential equation (PDE) Mémin & Pérez (2002), decomposing it into layers Wang & Adelson (1994); Weiss & Adelson (1996); Sun et al. (2010); Lao & Sundaramoorthi (2018), or grouping sparse trajectories Ochs et al. (2013); Keuper et al. (2015; 2016). The number of clusters/layers/objects was determined by regularization parameters or explicitly enforced, and these methods require solving complex optimization problems at inference time.

Segmentation models based on deep neural networks perform pixel-level classification into a number of classes equal to the number of output channels Noh et al. (2015); Zhao et al. (2017); Chen et al. (2017), which has to be decided ahead of time. Many "binary" methods Tokmakov et al. (2017); Jain et al. (2017); Song et al. (2018); Liu et al. (2021); Ding et al. (2022); Lian et al. (2023); Singh et al. (2023) are even more restrictive, considering two classes: "foreground" and "background" Hu et al. (2018); Fan et al. (2019); Li et al. (2019); Akhter et al. (2020). Some methods revisit layered models by supervising a DNN using synthetic data Xie et al. (2022) or use sprites Ye et al. (2022). However, a batch of frames must be processed before inference can be rendered on a given frame, and the model requires an explicit definition of the number of objects to be segmented, usually as a human input, rendering these approaches impractical in online and real-time scenarios. Meunier et al. (2022) segments optical flow by fitting pre-defined motion patterns (*e.g.*, affine, quadratic) using Expectation-Maximization (EM). Changing the number of outputs requires architectural modification and thus re-training the network. DivA is more flexible by only specifying an upper limit for the number of segments (slots) which can be conveniently adjusted without re-training the model.

Some recent motion segmentation methods Wang et al. (2022); Ponimatkin et al. (2023) use pre-trained image features from contrastive learning (e.g. DINO Caron et al. (2021)), while others Bao et al. (2022; 2023) rely on motion segmentation Dave et al. (2019) that leverages supervised features from MS-COCO Lin et al. (2014). There are also methods combining an unsupervised pipeline with existing supervised features Yang et al. (2021b); Haller et al. (2021). While in practice one ought to use pre-trained features whenever available, for academic purposes we explore the minimal end of the supervision scale, where no purposeful viewpoint selection, "primary object," or human labels are used. Thus we can bootstrap object discovery in a broader set of scenarios than those captured by the few multi-object discovery datasets available today. Given the efficiency of DivA, we hope in the future to test it in embodied settings, which current benchmarks are not representative of.

**Contextual Information Separation (CIS)** Yang et al. (2019) frames unsupervised motion segmentation as an information separation task. Assuming independence between motions as the defining trait of "objects," the method segments the optical flow field into two regions that are as uninformative of each other as possible. Although this formulation removes the notion of foreground and background it is still limited to binary partitioning. To the best of our knowledge, no work has yet extended the CIS method to multiple regions. Explicitly optimizing a multi-region version of CIS would result in a combinatorial explosion of complexity. Naive alternatives such as applying CIS sequentially to discover one object at a time did not yield satisfactory results. Instead, we exploit the architectural biases of SANs to bypass combinatorial optimization. **Slot Attention Networks (SANs)** Locatello et al. (2020) infer a set of latent variables each ideally representing an object in the image. Earlier "object-centric learning" methods Eslami et al. (2016); Greff et al. (2017); Kosiorek et al. (2018); Greff et al. (2019) aim to discover generative factors that correspond to parts or objects in the scene, but require solving an iterative optimization at test time; SAN processes the data in a feed-forward pass by leveraging the attention mechanism Vaswani et al. (2017) that allocates latent variables to a collection of permutation-invariant slots. SAN is trained to minimize the reconstruction loss, with no explicit mechanism enforcing the separation of objects. When objects have distinct features, the low-capacity bottleneck is sufficient to separate objects into slots. However, in realistic images, multiple objects tend to collapse to the same slots (Fig. 6), so a more explicit bias than that implicit in the architecture is needed. To that end, Kipf et al. (2021) resorts to external cues such as bounding boxes, while Elsayed et al. employs Lidar. Instead, DivA modifies the SAN architecture by using a cross-modal conditional decoder inspired by Yang & Soatto (2018). By combining infor-

---

[1]Optical flow can be inferred by optimization without supervision Horn & Schunck (1981). In the experiments we follow the convention Yang et al. (2019; 2021a); Bao et al. (2022; 2023); Meunier et al. (2022) to use RAFT Teed & Deng (2020) flow trained on synthetic data as an off-the-shelf component.

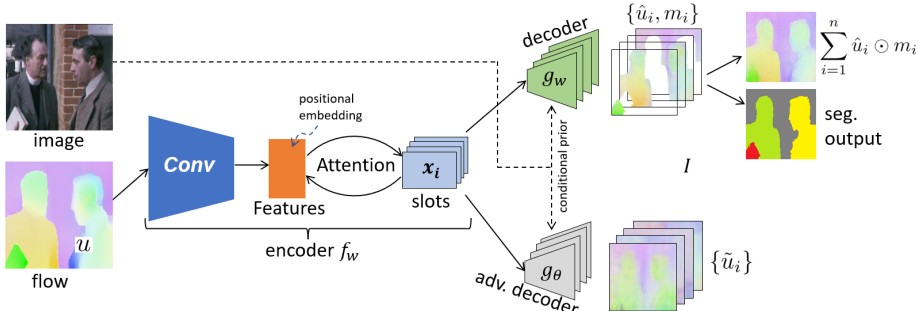

Figure 2: **Divided Attention Overview.** *Unlike traditional autoencoders that process the input (here the flow u) through an encoder $f_w$, and then reconstruct it through a decoder g, DivA uses a cross-modal conditional decoder ($g_w$ in green) that takes a second modality as input (here the image I) in order to reconstruct the first from the encoding "slots" $x_i$ (light blue). The image is used as a condition to guide the flow decoder, using a conditional prior network. To ensure that the individual slots encode separate objects, we incorporate an adversarial decoder ($g_\theta$ in grey) that tries to reconstruct the entire flow with each slot. Training is done by optimizing a min-max criterion whereby the model tries to reconstruct the input flow within each object mask, while fooling the adversarial decoder outside. This design enforces information separation between attention slots, leading to Divided Attention. The conditional decoder is expanded in Appendix Fig. 10.*

mation separation, architectural separation, and conditional generation, DivA fosters better partition of the slots into objects. DivA is most closely related to MoSeg Yang et al. (2021a), which adopts SAN for iterative binding to the flow. DivA has two key advantages: MoSeg relies on learned slots initializations (Fig. 3) and is limited to binary segmentation; MoSeg only uses flow while DivA employs a conditional cross-modal decoder guiding the segmentation mask to increased accuracy.

## 2 METHOD

DivA ingests a color (RGB) image $I \in \mathbb{R}^{H \times W \times 3}$ with H × W pixels and its associated optical flow $u \in \mathbb{R}^{H \times W \times 2}$ defined on the same lattice (image domain), and encoded as an RGB image using a color-wheel. DivA outputs a collection of $n$ masks $m_i$, $i = 1, \dots, n$ and the reconstructed flow $\hat{u}$.

### 2.1 PRELIMINARIES: SLOT ATTENTION AUTOENCODER

The DivA architecture comprises a collection of latent codes $X_n = \{x_1, \dots, x_n\}$, with each $x_i \in \mathbb{R}^{1 \times K}$ representing a "slot." The encoder $f_w(u) = X_n$ is the same as a Slot Attention Network (SAN), described in detail in Locatello et al. (2020) and summarized here. SANs are trained as autoencoders: An image $I$ is passed through a CNN backbone with an appended positional embedding, but instead of having a single latent vector, SAN uses a collection of them $\{x_i | i = 1, \cdots, n\}$, called *slots*, in the bottleneck, where $n$ may change anytime during training or testing. Slots are initially sampled from a Gaussian with learned mean and standard deviation, without conditioning additional variables. This affords changing the number of slots without re-training, and yields invariance to permutation the slots, which are updated iteratively using dot-product attention normalized over slots fostering competition among slots. The result is passed through a Gated Recurrent Unit (GRU) with a multi-layer perceptron (MLP) to yield the update residual for slots. All parameters are shared among slots to preserve permutation symmetry. Each slot is then decoded independently with a spatial broadcast decoder Watters et al. (2019) $g$ producing slot reconstructions $\hat{I}_i$'s and masks $m_i$'s. The final image reconstruction is $\hat{I} = \sum_{i=1}^{n} \hat{I}_i \odot m_i$ where $\odot$ denotes element-wise multiplication. SAN is trained by minimizing a reconstruction loss (typically MSE) between $I$ and $\hat{I}$.

### 2.2 CROSS-MODAL CONDITIONAL SLOT DECODER

Experimental evidence shows that slots can learn representations of independent simple objects on synthetic images. However, naive use of SAN to jointly auto-encode real-world images and flow leads to poor reconstruction. Since the combined input is complex and the slots are low-dimensional, slots tend to either approximate the entire input or segment the image in a grid pattern that ignores the objects. Both lead to poor separation of objects via the learned slots.

For these reasons, we choose *not* to use the image as input to be reconstructed, but as *context* to condition the reconstruction of a simpler modality – the one least affected by complex nuisance variability – flow in our case. The conditional decoder $g_w$ maps each latent code $x_i$ *and* image $I$ onto a reconstructed flow $\hat{u}_i = g_u(x_i, I)$ and a mask $m_i = g_m(x_i, I)$. With an abuse of notation, we write $g_w = (g_u, g_m)$ depending on whether we emphasize its dependency on the weights $w$ or on the component that generates the flow $u$ and the mask $m$ respectively.

This decoder performs cross-modal transfer since the image serves as a prior for generating the flow. This is akin to a Conditional Prior Network Yang & Soatto (2018), but instead of reconstructing the entire flow, we reconstruct individual flow components corresponding to objects in the scene, indicated by the masks. The flow is reconstructed by $\hat{u} = \sum_{i=1}^{n} \hat{u}_i \odot m_i$. Next, we further use an *adversarial conditional decoder* to generate $\tilde{u} = g_\theta(x_i, I)$, that attempts to reconstruct the entire flow $\tilde{u}$ *from each individual slot* $x_i$, which in turn encourages the separation between different slots.

## 2.3 ADVERSARIAL LEARNING WITH CONTEXTUAL INFORMATION SEPARATION

We now describe the separation criterion, which we derive from CIS Yang et al. (2019), to divide slots into objects. Each image $I$ and flow $u$ in the training set contribute a term in the loss:

$$\ell(u, I) = \left\| u - \sum_{i=1}^{n} \hat{u}_i \odot m_i \right\| - \frac{\lambda}{n} \sum_{i=1}^{n} (1 - m_i) \odot \| u - \tilde{u}_i \| \qquad (1)$$

The first term penalizes the reconstruction error by the cross-modal autoencoder combining all the slots, i.e., we want the reconstructed flow to be as good as possible. The second term combines the reconstruction error of the adversarial decoder using a single slot at a time, and maximizes its error outside the object mask. Note that, in the second term, the adversarial decoder $g_\theta$ tries to approximate the entire flow $u$ with a single slot $\tilde{u}_i = g_\theta(x_i, I)$, which is equivalent to maximizing the contextual information separation between different slots. Also note that we use the mean-square reconstruction error (MSE) $d(u, v) = \| u - v \|_2$ for simplicity, but one can replace MSE with any other distance or discrepancy measure such as empirical cross-entropy, without altering the core method. The overall loss, averaged over training samples in a dataset $D$, is then minimized with respect to the parameters $w$ of the encoder $f_w$ and conditional decoder $g_w$, and maximized with respect to the parameters $\theta$ of the adversarial decoder $g_\theta$:

$$\min_{w} \max_{\theta} L(w, \theta) = \sum_{(u_j, I_j) \in D} \ell(u_j, I_j). \qquad (2)$$

DivA is minimizing mutual information *between different slots*, where the data is encoded, rather than directly *between different regions*, which eliminates degenerate slots thanks to the reconstruction objective. The resulting design is a natural extension of CIS to non-binary partitions, and also leads to increased versatility: DivA can be trained with a certain number of slots, on images of a certain size, and used at inference time with a different number of slots, on images of different size. Note that the loss does not need explicit regularization, although one can add it if so desired.

## 3 EXPERIMENTS

**Encoder.** The encoder $f$ consists of 4 convolutional layers with kernel size equal to $5 \times 5$, consistent with the original implementation of Locatello et al. (2020). With padding, the spatial dimension of the feature map is the same as the network input. Since optical flow is simpler to encode than RGB images, we choose $K = 48$ instead of 64 used by the original SAN, resulting in a narrower bottleneck. A learned 48-dimensional positional embedding is then added to the feature map. We tried Fourier positional embedding and the network shows similar behavior. We keep the iterative update of slots the same as the original SAN, and fix the number of slot iterations to be 3.

**Conditional decoder and adversarial decoder.** The architecture of the conditional decoder $g_w$, shown in Appendix Fig. 10, consists of two parts: an image encoder, and a flow decoder. We use 5 convolutional layers in the image encoder. Same as $f$, with padding, the size of the feature maps remains the same as $I$. We limit the capacity of this encoder by setting the output channel dimension of each layer to 24 to avoid overfitting the flow to the image, ignoring information from the slots.

| Method | Resolution | Multi | DAVIS | ST | FPS |
|---|---|---|---|---|---|
| Unsupervised Single-frame Method | | | | | |
| CIS Yang et al. (2019)(4) | $128 \times 224$ | N | 59.2 | 45.6 | 10 |
| CIS(4)+CRF | $128 \times 224$ | N | 71.5 | 62.0 | 0.09 |
| MoSegYang et al. (2021a) | $128 \times 224$ | N | 65.6 | - | 78 |
| MoSeg(4) | $128 \times 224$ | N | 68.3 | 58.6 | 20 |
| EM* Meunier et al. (2022) | $128 \times 224$ | Y* | 69.3 | 60.4 | 21 |
| DivA | $128 \times 128$ | Y | 68.6 | 60.3 | **104** |
| DivA | $128 \times 224$ | Y | 70.8 | 60.1 | 64 |
| DivA-Recursive | $128 \times 224$ | Y | 71.0 | 60.9 | 66 |
| DivA(4) | $128 \times 224$ | Y | **72.4** | **64.6** | 16 |
| Video Batch Method | | | | | |
| IKE Haller et al. (2021) | $224 \times 416$ | N | 70.8 | 66.4 | - |
| USTN Meunier & Bouthemy (2023) | $128 \times 224$ | Y | 73.2 | 55.0 | - |
| Deformable Sprites Ye et al. (2022) | Full | Y | 79.1 | 72.1 | - |
| With Additional Supervision | | | | | |
| FSEG Jain et al. (2017) | Full | N | 70.7 | 61.4 | - |
| OCLR Xie et al. (2022) | $128 \times 224$ | Y | 72.1 | 67.6 | - |
| ARP Koh & Kim (2017) | Full | N | 76.2 | 57.2 | 0.015 |
| DyStab Yang et al. (2021b) | Full | N | 80.0 | 73.2 | - |

Table 1: **Binary Moving object segmentation results.** *Although not exploiting the prior of a dominant "foreground object", DivA still achieves compelling accuracy on binary segmentation datasets, with better inference speed. As a single-frame method that requires no pre-trained features, DivA closes the gap to methods that require video batch processing (therefore cannot be deployed to online and real-time applications) and methods using additional supervision or pre-trained features. Multi: method generalizes to multi-object segmentation.*

The flow decoder takes one $1 \times 48$ slot vector $x_i$ as input. It first broadcasts $x_i$ spatially to $h \times w \times 48$, and adds it to a learned positional embedding. Note that the positional embedding is different from the one in $f$. The broadcasted slot then passes through 6 convolutional layers. The feature map at each layer is concatenated with the image feature at the corresponding level. The last convolutional layer outputs a $h \times w \times 4$ field, where the first 3 channels reconstruct the optical flow, and the 4th channel outputs a mask. The adversarial decoder shares the same architecture, except for the last layer, which outputs 3 channels aiming to reconstruct the flow on the entire image domain.

**Optimization.** We aim to keep the training pipeline simple and all models are trained on a single Nvidia 1080Ti GPU with PyTorch. We implement implicit differentiation during training Chang et al. (2022) for stability, and apply alternating optimization to $w$ and $\theta$ by Eq. equation 2. In each iteration, we first fix $g_\theta$ and update $w$, then use *torch.detach()* to stop gradient computation on $x_i$ before updating $\theta$. This makes sure that only $\theta$ is updated when training the adversarial decoder. We train with batch size 32 in all experiments and apply the ADAM optimizer with an initial learning rate $8e^{-4}$ for both $w$ and $\theta$. We notice that the model is not sensitive to the choice of learning rate. We use RAFT for optical flow in all experiments unless specified. More details are in Appendix A.

## 3.1 BINARY SEGMENTATION

While DivA is designed to segment multiple moving objects, we first report results on binary segmentation, a special case where $n = 2$, due to the scarcity of benchmarks and baselines for multiple moving object segmentation. **DAVIS2016** Perazzi et al. (2016) consists of 50 videos, each ranging from 25 to 100 frames containing one primary moving object that is annotated. **SegtrackV2** Tsai et al. (2012) has 14 video sequences with a total of 947 frames with per-frame annotation. Annotated objects in the video have apparent motion relative to the background.

We compare with: **CIS** Yang et al. (2019) uses a conventional binary segmentation architecture. It employs an auxiliary flow inpainting network and enforces information separation by minimizing mutual inpainting accuracy on the regions. **Moseg** Yang et al. (2021a) adopts SAN for iterative binding to the flow field, fixes the number of slots to 2, and learns slot initialization instead of using random Gaussian. It is trained by reconstruction loss together with temporal consistency. **EM** Meunier et al. (2022) pre-defines a class of target motion patterns (*e.g.,* affine), and alternately updates segmentation and corresponding motion patterns during training. We select them as baselines since they 1) use single-frame (instantaneous motion) without exploiting cross-frame data association; 2) do not rely on the presence of a dominant "foreground object". DivA is chosen so as to be efficient and flexible, leading to 1), and not biased towards purposefully framed videos, leading to 2).

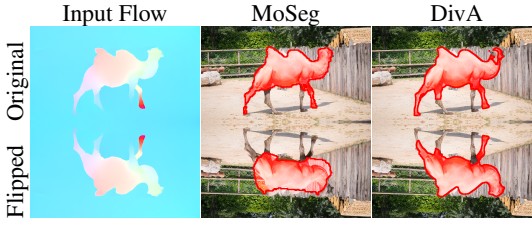
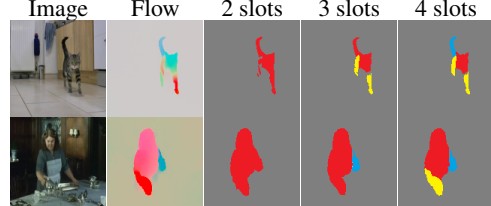

Figure 3: **Conditional decoder is less vulnerable to overfitting.** *Reconstructing complex input from slots forces the decoder to overfit to seen data. Simply flipping the input flow drastically decreases segmentation accuracy. By introducing the conditional decoder, DivA is less vulnerable to overfitting.*

Figure 4: **Varying the number of slots changes the granularity of segmentation.** *All the above results are obtained from the same model by only varying the number of slots at inference time, without re-training. This gives users additional control over the granularity of segmentation.*

Results are summarized in Table 1. Our best result outperforms the best-performing baseline EM by 4.5% and 6.9%, respectively. We measure the run time of one single forward pass of the model. Our fastest setting using $128 \times 128$ resolution reaches 104 frames per second. In addition to running on consecutive frames, CIS and MoSeg merge segmentation results on optical mapping from the reference frame to 4 adjacent temporal frames, marked by "(4)" in the table. The same protocol improves DivA on both datasets. Compared to models using additional supervision, our best performance exceeds the FSEG Jain et al. (2017) (supervised) and OCLR Xie et al. (2022) (supervised, trained on synthetic data), and closes the gap with the best-performing method DyStab Yang et al. (2021b) to 12%. Inspired by Elsayed et al., we test DivA by recursively inheriting slots instead of randomly initializing them, and reducing the number of iterations from 3 to 1. Both segmentation accuracy and runtime speed improve marginally. The base model randomly permutes the labels; with recursion, label-flipping is reduced. Unlike MoSeg and DyStab which employ explicit regularizations, our method exhibits temporal consistency without any modification to the training pipeline.

Furthermore, we compare with MoSeg which also uses slot attention. Without the conditional decoder, due to the limited capacity of the slots, MoSeg relies on the decoder to reconstruct fine structures of the input flow, forcing the decoder to memorize shape information. Together with learned slot initializations, the model is subject to overfitting. Fig. 3 shows an example, where simply flipping the input leads to performance degradation. Conditioning the decoder on the image frees it from memorizing the flow to be reconstructed, thus making the model less vulnerable to overfitting.

## 3.2 MULTIPLE REGION SEGMENTATION

**Diagnostic data and ablation.** Due to the limited availability of benchmark datasets for multiple moving objects segmentation, we reproduce the ideal conditions of the experiments reported in Yang et al. (2019) to perform controlled ablation studies. We generate flow-image pairs that contain $n = 2, 3, 4$ regions with statistically independent motion patterns. We adopt object masks from DAVIS and paste them onto complex background images so that the network cannot overfit to image appearance for reconstruction and segmentation. During training, we fix the number of slots to be 4 and the network is unaware of the number of objects. Examples are in the Appendix (Figure 11).

We evaluate on 300 validation samples and measure the performance by *bootstrapping IoU (bIoU)* and *successful partition counts (SPC)*. bIoU is computed by matching each ground truth object mask to the segment with the highest IoU, and averaging this IoU across all objects in the dataset. It measures how successfully each object is bootstrapped. As bIoU does not penalize falsely bootstrapped blobs, in addition, SPC counts the number of cases where the number of objects in the segmentation output matches the ground truth number of objects. As the architecture itself is not aware of the number of objects in each flow-image pair during testing, multiple slots may map to the same object. A higher SPC is achieved when information is separated among slots, reducing such confusion.

We apply SAN to *reconstruct the flow* as the baseline [2], and also combining SAN with the conditional decoder. Table 2 summarizes the results. Our conditional decoder allows exploiting photometric information, in addition to motion, improving bIoU; adversarial learning fosters better separation among slots, reducing slot confusion and improving SPC. In Fig. 5 we display the scatter between

---

[2] CIS, MoSeg are binary segmentation, and EM fixes the number of outputs, not amenable to this task.

| Method | bIoU | SPC |
|---|---|---|
| Vanilla SAN | 51.18 | 120 |
| SAN + Cond. Decoder | 82.93 | 133 |
| DivA ($\lambda = 0.01$) | **84.49** | 161 |
| DivA ($\lambda = 0.03$) | 82.33 | **182** |
| DivA ($\lambda = 0.05$) | 79.56 | 174 |

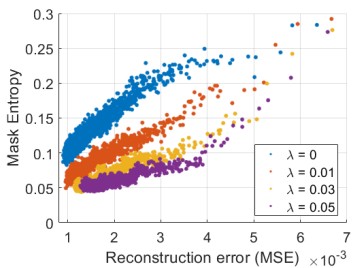

Table 2: **Results on diagnostic data.** *Conditional decoding significantly improves bootstrapping IoU (bIoU), indicating better awareness of object shape; adversarial training significantly improves successful partition count (SPC), indicating better information separation in slots.*

Figure 5: **Adversarial loss is an implicit mask entropy regularizer.** *At each level of reconstruction error, the higher $\lambda$ we apply, the smaller entropy we get in the segmentation output. With the adversarial loss, the model successfully predicts the number of independent motion patterns on more samples.*

| | $\delta t = 1$ | | | $\delta t = 2$ | | |
|---|---|---|---|---|---|---|
| n | SAN | w/ cond. | DivA | SAN | w/ cond. | DivA |
| 3 | 17.76 | 39.81 | 40.87 | 17.66 | 40.90 | 41.27 |
| 4 | 19.27 | 40.96 | **42.54** | 19.57 | 39.81 | **41.85** |
| 5 | 20.17 | 38.73 | 40.25 | 20.15 | 38.47 | 40.48 |
| 6 | 19.69 | 38.03 | 38.90 | 17.90 | 38.59 | 40.26 |

Table 3: **Bootstrapping accuracy on FBMS-59.** *We evaluate the performance of DivA on FBMS-59 by bIoU, varying the number of slots in the network (without re-training). We report results on the baseline SAN Locatello et al. (2020), SAN with our conditional decoder, and the full DivA model.*

reconstruction error and segmentation Mask Entropy $= \sum_i^4 m_i \log(m_i)$ during training. A smaller entropy corresponds to a more certain mask output, indicating that the decoder relies more on single slots to reconstruct optical flow for each particular object. At each level of reconstruction error, the larger $\lambda$, the smaller the mask entropy. This phenomenon validates the effect of our adversarial training. Note that entropy regularization is also applied to existing unsupervised segmentation methods (*e.g.* MoSeg), and our adversarial loss provides a potential alternative to entropy regularization.

**Results on FBMS-59.** FBMS-59 dataset contains 59 videos ranging from 19 to 800 frames. In the test set, 69 objects are labeled in the 30 videos. We train with $n = 4$ aiming to extract *multiple* independently moving objects from a scene. At test time, we vary $n$ ranging from 3 to 6 without re-training the model, and report bootstrapping IoU. We test the model on optical flow computed with $\delta t = 1$ and $\delta t = 2$, and keep the spatial resolution to $128 \times 128$ in both training and testing.

Table 3 summarizes the results. The conditional decoder improves segmentation accuracy substantially, and the adversarial training further improves accuracy by 3.9% in the best-performing case. We notice that $n = 4$ is a sweet spot for the dataset, due to the dataset's characteristic of having less than 3 moving objects in most sequences. Fig. 1 shows qualitative examples. For many objects segmented, we observe a change in the level of granularity of the segmentation mask when varying $n$. Two examples are given in Fig. 4. Although depending on the annotation, both cases may be considered as over-segmenting the object in quantitative evaluations and degrade IoU, we argue that such a feature is desirable. As DivA can conveniently vary $n$ without re-training, it reveals DivA's potential in developing adaptive, interactive, or hierarchical segmentation in future work.

**Quantitative results on Movi-Solid.** [3] Movi-Solid Singh et al. (2022) is a realistic synthetic dataset containing 9000 sequences featuring various, unidentified numbers of objects. Most objects exhibit motion relative to the dynamic background. We train DivA on $128 \times 128$ resolution with 4 slots. At test time, we increase the number of slots to $n = 10$. As in Fig. 6, DivA yields compelling segmentation results. The SAN model, although trained with the same reconstruction loss, fails to capture object boundaries. When equipped with a conditional decoder, SAN yields better mask alignment with object boundaries but tends to over-segment due to the absence of an information separation criterion. For completeness, we also train SAN on image reconstruction denoted as "SAN-img". Given the complexity of nuisance in the dataset, it fails to yield meaningful segmentation. This underscores the significance of using motion as a robust cue for unsupervised object discovery.

---

[3]Ground-truth segmentation masks of Movi-Solid are not publicly available at the time of paper submission.

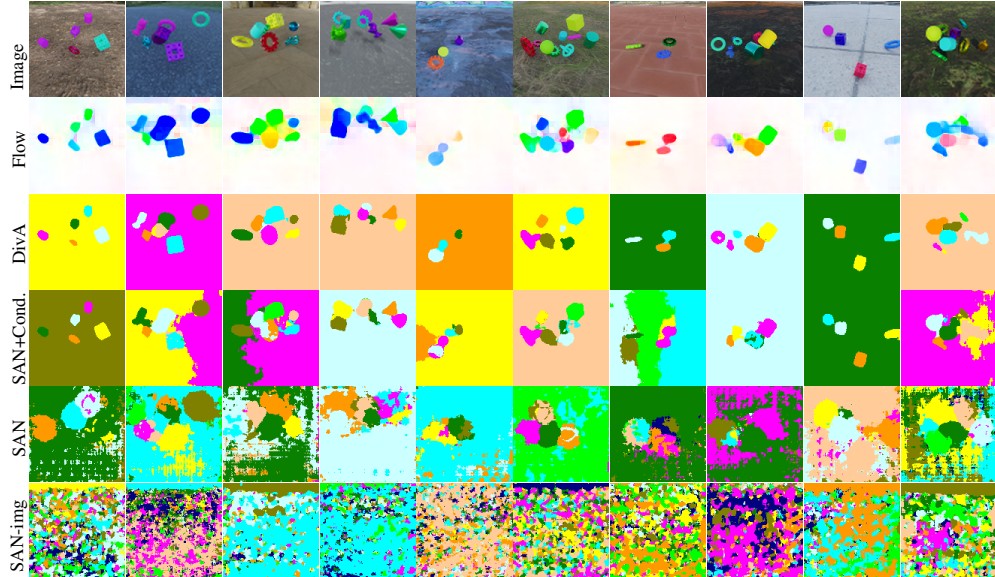

**Figure 6: Quantitative results on Movi-Solid.** *DivA outputs compelling segmentation, while SAN struggles with object boundaries. Adding a conditional decoder to SAN improves boundary alignment but is still vulnerable to over-segmentation. Naively applying SAN for image reconstruction (SAN-img) yields non-informative results, emphasizing the importance of motion cues in unsupervised object discovery.*

## 4 DISCUSSION

DivA is a multi-object discovery model trained to perform contextual information separation and cross-modal validation, with no manual supervision. The architecture is based on Slot Attention, but modified to comprise a cross-modal decoder, turning a simple autoencoder into a conditional encoder, and an adversarial decoder. The loss function uses the adversarial decoder to enforce Contextual Information Separation *not* directly in the data, but in the latent space of the slot encodings. The overall system enjoys certain properties not observed in existing methods: *First*, it does not require specifying the number of objects, and allows for their upper bound to change between training and test time. *Second*, the labels are permutation invariant, and DivA can be used in a recursive fashion to reduce label flipping by simply using slots from previous frames as initialization for incoming frames. Other methods for doing so Elsayed et al. require manual input and modification of the training pipeline or require temporal consistency heuristics Yang et al. (2021b;a). *Third*, the model can be trained on images of a certain resolution and then used on different resolutions at inference time. We have trained on $128 \times 128$ images to spare GPU memory and test on $128 \times 224$ for accuracy, and vice-versa one can train on larger images and use at reduced resolution for speed. *Fourth*, DivA does not require sophisticated initialization: It operates using randomly initialized slots and is simply trained on image-flow pairs. It is trained with 4 slots, and tested with 3, 4, 5 and 6 slots, capturing most of the objects in common benchmarks. *Fifth*, DivA is fast. *Finally*, DivA is a generative model. It can be used to generate segmentation masks at high resolution, without post-processing (*e.g.* CRF) to refine the results, by directly upsampling the decoded slots to the original resolution.

DivA has several limitations. While we have tested DivA on a handful of objects in benchmark datasets, scaling to hundreds or more slots is yet untested. There are many variants possible to the general architecture of Fig. 2, including using Transformer models (e.g. Singh et al. (2021; 2022)) in lieu of the conditional prior network. With more powerful multi-modal decoders, the role of motion may become diminished, but the question of how the large model is trained (currently with aligned visual-textual pairs), remains. Since our goal is to understand the problem of object discovery *ab ovo*, we keep our models minimalistic to be trained efficiently even if they do not incorporate the rich semantics of natural language or human-generated annotations. Common failure modes of DivA are analyzed in the Appendix Fig. 8. We can envision several extensions of DivA. For example, even though the amphibian visual system requires objects to move in order to spot them, primate vision can easily detect and describe objects in static images, thus, a bootstrapping strategy can be built from DivA. Moreover, varying the number of slots changes the level of granularity of the slots (Fig. 4), which leads to the natural question of how to extend DivA to hierarchical partitions.

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

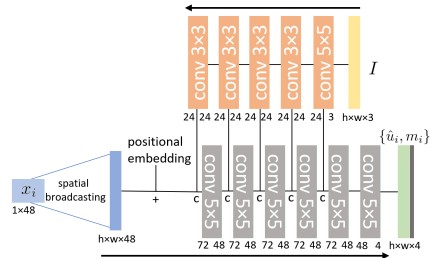

Figure 7: **Architecture of the conditional decoder.**

# A  IMPLEMENTATION DETAILS

**Training and testing.**  We apply RAFT Teed & Deng (2020) for optical flow following Yang et al. (2021a); Meunier et al. (2022). For image sequence $\{I_t\}$, RAFT computes optical flow from $I_t$ to $I_{t+\delta t}$, where $\delta t$ is sampled from -2 to 2. Optical flow then downsampled together with the reference image before feeding to the network. During training, we warm up our model on DAVIS2016 dataset, setting $n = 4$. During this warm-up stage, we set $\lambda = 0$ so that the model can learn reconstruction in the initial stage without the interference of an adversarial decoder. After the warm-up, we train the model on each particular dataset, with $n = 2$ on DAVIS2016 and SegtrackV2, and $n = 4$ on FBMS-59. Referring to the empirical evidence on the diagnostic dataset, We set $\lambda = 0.03$, and decrease it to $\lambda - 0.01$ towards the end of the training. We use a batch size of 32 and set the spatial resolution of model input to be $128 \times 128$ when $n = 4$ and $128 \times 224$ when $n = 2$, tailored by the GPU's memory constraint.

At test time, we keep $n = 2$ on DAVIS2016 and SegtrackV2, and vary $n$ on FBMS-59. On binary segmentation, since the model is trained without the notion of foreground and background, we follow the baselines to match the ground truth with the most likely segmentation mask and compute intersection-over-union (IoU) to measure the segmentation accuracy. We upsample the segmentation output to the dataset's original resolution for evaluation. Unlike baseline methods that upsample low-resolution segmentation masks by interpolation, our generative model reconstructs flow from each slot, so we can refine the segmentation outputs by upsampling $\{\hat{u}_i\}$'s to the full resolution, then refine the segmentation boundaries by $\arg\min_i |u - \hat{u}_i|_2$. This practice creates negligible computational overhead (0.00015s) and empirically improves segmentation accuracy. On multi-object bootstrapping, we follow the bootstrapping IoU defined in Sect. 3.2 and evaluate the accuracy per instance. Note that this is different from the baseline methods that merge all instances into a single foreground for evaluation.

**Data Normalization.**  DivA takes optical flow and image as input. Since there is no restrictions on the magnitude of the flow, we first convert the 2-channel flow field to an RGB image by a standard color-wheel, similar to CIS Yang et al. (2019) and MoSeg Yang et al. (2021a). This practice serves as a data normalization that is robust to the magnitude of optical flow. During both training and testing, we then normalize the input RGB flow to zero-mean and range in [-1,1]. We notice that normalizing the flow to zero-mean, which eliminates a global translation in the motion field, leads to more stable training. The input image is also normalized to within [-1,1], similar to SAN Locatello et al. (2020).

**Training Parameters.**  For $w$ and $\theta$ we train with two separate ADAM optimizers to perform alternating optimization. In the warm-up stage, $w$ is trained with $\lambda = 0$ (no adversarial loss). Note that even if the adversarial loss is not applied, we still update $\theta$ at this stage, so the adversarial decoder still learns to reconstruct the flow on the whole image domain from each slot. Both optimizers use an initial learning rate of $8e^{-4}$ with linear learning rate decay. On diagnostic data, we train for 1000 epochs. On real-world data, the network first warm-ups for 200 epochs on DAVIS with $n = 4$ and spatial resolution $128 \times 128$, then train on each dataset under settings specified in Sect. 3.2 in the main paper for different experiments. This warm-up step provides a stable initialization to the network, especially when the target dataset is small (e.g. SegTrackV2). On DAVIS and SegTrackV2 the network is trained for 500 epochs, and on FBMS-59 the network is trained for 200 epochs since the dataset is larger.

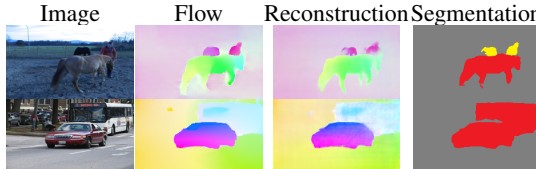

Figure 8: **Failure modes.** *DivA is trained using the CIS principle, which assumes objects move independently. Consequently, two objects moving in the same way may share high mutual information, e.g., even though the car and the bus are different detached objects, if their motions are highly correlated in the current video, they are seen as one by DivA (of course, as time goes by, their motions will diverge, allowing DivA to correctly separate them).*

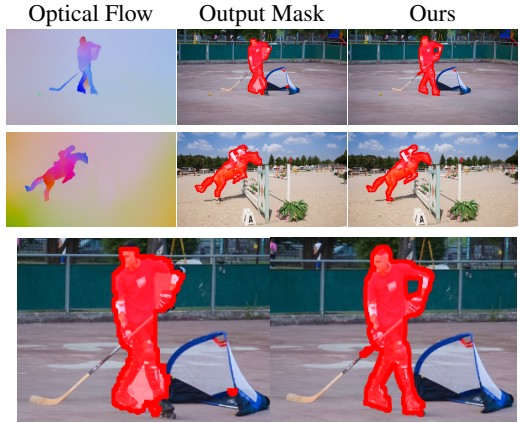

Figure 9: **Upsampling segmentation masks to full resolution.** *Better viewed zoomed-in. Our generative model allows upsampling segmentation masks to full resolution without losing details of object boundaries. Details for the upper panel are shown side-by-side in the last row.*

## B  ADDITIONAL QUALITATIVE TESTS

**Upsampling segmentation masks to full resolution.** Many moving object segmentation techniques rely on low-resolution input data Yang et al. (2019; 2021a); Meunier et al. (2022), and output segmentation masks are then upsampled through interpolation to achieve full resolution. However, this interpolation process leads to a loss of detail, especially along object boundaries. In contrast, the DivA generative model offers an alternative approach to upsampling that involves comparing upsampled flow reconstruction with the full-resolution optical flow by $\arg\min_i |u - \hat{u}_i|_2$, thereby refining the segmentation boundaries without significantly increasing computational overhead. This allows for the segmentation masks that match the optical flow at full resolution, producing smoother segmentation boundaries that align well with the moving objects in the image. Figure 9 provides representative samples of the outcome of this upsampling procedure (better visible after zooming-in). Compared with directly upsampling the output mask (second column), our approach (third column) captures fine details of object boundaries without the artifacts that are typically associated with upsampling.

**Conditional Decoder.** The DivA model employs a conditional decoder that utilizes an image as a prior to reconstructing optical flow from slots. Two examples of the reconstructed optical flow are presented in Figure 10. In these examples, the input flow misses the fine structure of the object due to the assumptions underlying flow estimation, which causes artifact in the resulting flow estimate. Despite the flaws of the input flow, the reconstruction through the conditional decoder correctly captures the object boundaries. This illustrates the fact that the decoder learns to reconstruct the flow by leveraging photometric information such as high-contrast edges, their shape, and more general photometric structures in the image. In the example in the figure, the conditional prior helps the model troubleshoot the "mono-leg" in the middle column. This opens up the potential of bootstrapping objects in static images by motion directly from the DivA model.

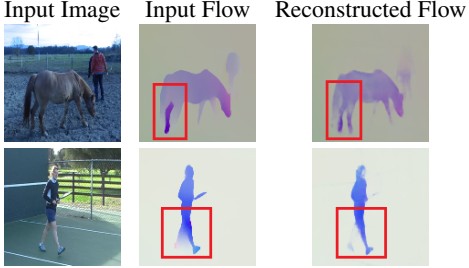

Figure 10: **Conditional decoder learns image priors.** *Despite imperfections in the input optical flow within the highlighted region, our reconstructed flow exhibits a satisfactory alignment with object boundaries. This finding indicates that the conditional decoder effectively employs image priors for flow reconstruction.*

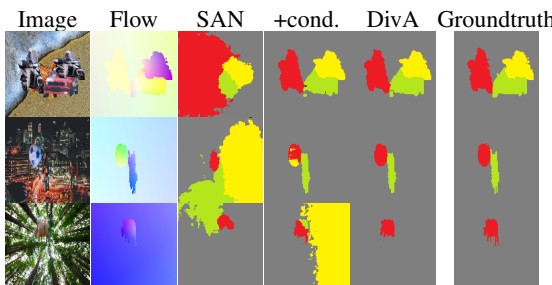

Figure 11: **Representative results on diagnostic data.** *Vanilla slot attention model can roughly capture moving objects but fails to output accurate segmentation. With conditional decoder, masks align better to object boundaries, but multiple slots may map to the same object. Incorporating adversarial training promotes greater information separation between slots.*

**Examples for Diagnostic Data.** Figure 11 shows representative examples in the diagnostic data. We utilize objects and corresponding segmentation masks extracted from the DAVIS dataset and merged them with arbitrary background images. Objects (including the background) are assigned motion patterns that are statistically independent. An image may contain $n = 2, 3, 4$ independent motion patterns and the network is unaware of the exact number. We use complex natural images so that the network cannot overfit to simply segmenting RGB images.

On our diagnostic dataset, the vanilla slot attention model fails to output accurate segmentation. With the introduction of a conditional decoder, masks align better to object boundaries, but multiple slots may map to the same motion. The full DivA model further promotes information separation between slots, especially when the number of slots exceeded the number of independent motions, resulting in more accurate segmentation outputs.

## C   ADDITIONAL DISCUSSION

**Empty Slots.** In cases where the number of slots is more than the number of independent motion patterns in the input, certain slots may remain unallocated to any region in the image domain. These slots do not carry any information about the input flow, and thus the flow reconstructed from these slots are random patterns and the corresponding masks are close to zero. Future work will focus on dynamically allocating the number of slots so that they adapt to the input data, especially in videos where objects appear and disappear as a result of occlusions.

**Recursive Implementation.** We provide a recursive implementation of the algorithm in which slots within the current frame are initialized with corresponding slots from prior frames, and updated by 1 iteration instead of 3. This approach makes the model less vulnerable to label-flipping provided the motion pattern is consistent across frames. However, when the motion undergoes drastic changes, for example when an object stops moving suddenly and then resumes after a hiatus, the model fails to associate the object to the same slot. In order to address this issue, future work on dynamic management of slots will also address such long-term consistency issues.

| Image Sequence | Optical Flow | Segmentation |
| --- | --- | --- |

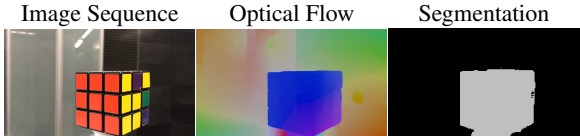

Figure 12: **Segmenting a static scene.** *In a static scene, changes in viewpoint can lead to a piecewise smooth optical flow, which can be segmented by DivA.*

**Static scene.** In this work, our goal is to discover objects based on motion signals. One question arises: can the proposed method effectively find objects in static scenes? To address this, we present an example in Figure 12. Even in a static scene, changes in viewpoint can lead to a piecewise smooth optical flow, which our method, DivA, can segment. According to Gibson's concept of "detached objects" as "layouts of surfaces surrounded by the medium," camera motion can create occlusion boundaries in any scene that is not globally convex. As a result, one moving agent can observe piecewise smooth optical flow with occluding boundaries, induced by either viewer or object motion. In such scenarios, DivA demonstrates its ability to identify objects when embedded in a moving agent, such as a robot, even in a static scene.

**Negative Results.** The main paper shows that adversarial training in DivA functions as an implicit entropy regularization. However, our attempt to integrate the entropy loss directly into the overall loss

$$\ell_{\text{entropy}}(u, I) = \sum_{i=1}^{n} \big\| m_i \log(m_i) \big\| \tag{3}$$

is actually detrimental of overall performance. We speculate that the entropy loss solely stimulates binary segmentation output and does not mandate any form of information separation. Incorporating this loss may keep the model from capturing the motion pattern of the complete object. Additional work will be needed to analyze the regularizing role of the adversarial loss and compare it to explicit regularization forms that are believed to foster information separation of "disentanglement."

