# OpenReview forum: "Divided Attention: Unsupervised Multiple-object Discovery and Segmentation with Interpretable Contextually Separated Slots"
_ICLR.cc/2024/Conference — Submitted to ICLR 2024_

### Official Review · Reviewer_5Uwa · 2023-10-29

**Soundness:** 3 good
**Presentation:** 3 good
**Contribution:** 3 good
**Rating:** 5
**Confidence:** 3

**Summary:**

This paper proposed a motion segmentation method to segment multiple objects, based on optical flow in an unsupervised manner. In this setting, both images and optical flow are available. Based on the SAN method, this work proposed an adversarial conditional encoder-decoder architecture. The proposed method can handle different numbers of objects at training and test time. The experimental results demonstrate the effectiveness of the proposed method.

**Strengths:**

(1) It can handle different numbers of objects at both training and test time.

(2) It can run in real-time.

(3) The performance is good.

**Weaknesses:**

(1) The ability of handling multi-object comes from SAN. The main contribution may be the adversarial framework and the manner using the image information. The authors should highlight the contributions of this paper.

(2) Handling multi-object is not new in video object segmentation.

(3) Why g_theta is an "adversarial" decoder is not clear. It forces the decoder to reconstruct the entire flow with each slot, which seems the two decoder do not have an adversarial relationship.

(4) This method "frees the representation from having to encode complex nuisance variability in the image", which should be demonstrated in the experiments. For example, simply combining (concat) the image and the optical flow as input can be considered as a baseline. Although the authors mentioned combined input is complex and the slots are low-dimentional. More reasonable explanation is needed.

(5) This setting is interesting,  but I guess its performance is heavily related to performance of the optical flow network. In the inference stage, the optical flow is also need to calculate first.

(6) P4, DivA has two key advantages, but the following text is mainly about the disadvantages of MoSeg.

**Questions:**

see the weaknesses.

---

> ### Author Response · Authors · 2023-11-16
> **Author response to the initial review**
>
> Thank you for your thoughtful review. Here we address your concerns below:
>
> **W1:** *The ability of handling multi-object comes from SAN. The main contribution may be the adversarial framework and the manner using the image information. The authors should highlight the contributions of this paper.*
>
> **R1:** We appreciate your suggestion and will make revisions to highlight the contributions.
>
> Regarding SAN, it's important to note that, although SAN is designed to handle multi-object, naively applying SANs to motion leads to disappointing results, which is validated by the results in Table 2, Table 3, and Figure 5. To make SANs effective when applied to motion, our key contributions include 1) introducing a conditional decoder to align the segmentation mask with object boundaries and 2) incorporating an adversarial framework to render the slots in SAN mutually uninformative. We will make them clear to the readers.
>
> **W2:** *Handling multi-object is not new in video object segmentation.*
>
> **R2:** We acknowledge that we are not the first to address multiple objects, and do provide a detailed literature review of multi-object segmentation in Sect. 1.1. However, our method, DivA, stands out as the first to exhibit several distinctive features:
> - It can handle a different number of objects without the need for re-training (so that it is versatile for real applications where the number of object is unknown).
> - It trains without explicit regularization (so that it can be trained on any generic video data).
> - It only requires single image-flow pairs and operates in real-time (so that it is suitable for deployment in close-loop real-time perception systems with low algorithmic latency).
> - It does not necessitate pre-training (so that it discovers "objects" from scratch and is not susceptible to data bias in pre-training).
>
> These unique characteristics demonstrate its advantages over existing multi-object methods.
>
> **W3:** *Why $g_\theta$ is an "adversarial" decoder is not clear. It forces the decoder to reconstruct the entire flow with each slot, which seems the two decoder do not have an adversarial relationship.*
>
> **R3:** The adversarial relationship arises when $g_\theta$ aims to reconstruct the entire flow, while the $f_w$ aims to hinder $g_\theta$ from reconstructing the flow OUTSIDE the region $m_i$ from $x_i$, so that each slot $x_i$ is only "informative" about the flow inside $m_i$, and is "uninformative" about the flow outside $m_i$. This is modeled by the mini-max formulation of the loss in equation (2), and minimizes mutual information between different slots. Please refer to Sect. 2.3 for the details.
>
> **W4:** *This method "frees the representation from having to encode complex nuisance variability in the image", which should be demonstrated in the experiments. For example, simply combining (concat) the image and the optical flow as input can be considered as a baseline. Although the authors mentioned combined input is complex and the slots are low-dimentional. More reasonable explanation is needed.*
>
> **R4:** This is a good advice. In fact, we have tried this setup by concatenating the image to the flow and having SAN reconstruct the flow. On page 4, Sect 2.2, we wrote "However, naive use of SAN to jointly auto-encode real-world images and flow leads to poor reconstruction." This refers to the exact setting you are suggesting, and we will revise the paper to make it clear.
>
> The reason is, that although the input consists of both the image and flow, they are compressed through a narrow bottleneck. Due to nuisance variability, reconstructing from this highly compressed slot poses a challenge for the decoder, especially in real-world videos where the objects have complex shapes and textures. Our conditional decoder addresses this issue by allowing the slot to provide a general "hint" for reconstruction, with the conditional decoder handling fine details. Please refer to Figure 10 in the Appendix for a more detailed discussion.
>
> **W5:** *This setting is interesting, but I guess its performance is heavily related to the performance of the optical flow network. In the inference stage, the optical flow is also needed to calculate first.*
>
> **R5:** We agree that this is a legitimate concern, and applies to all object discovery schemes based on optical flow. And that's indeed why we keep the optical flow consistent with baseline methods (MoSeg, EM, etc.) when making comparisons.
>
> **W6:** *P4, DivA has two key advantages, but the following text is mainly about the disadvantages of MoSeg.*
>
> **R6:** We will rephrase.
>
> We hope this response answers the reviewer's question. If there are further concerns, we are more than willing to provide additional explanations.

---

### Official Review · Reviewer_RqAo · 2023-11-01

**Soundness:** 3 good
**Presentation:** 3 good
**Contribution:** 3 good
**Rating:** 6
**Confidence:** 2

**Summary:**

This paper proposes a method to segment the visual field into independently moving regions. The proposed method uses a cross-modal conditional decoder that takes a second modality as input  to reconstruct the first modality.  This design frees the representation from having to encode complex nuisance variability in the image, such as illumination and reflectance properties of the scenes.

**Strengths:**

This paper is well-written and easy to follow.

The experimental results demonstrate better performance.

The idea is simple and effective.

**Weaknesses:**

I cannot find the obvious weakness.

**Questions:**

How about the GPU memory consumption?

---

> ### Author Response · Authors · 2023-11-16
> **Author response to the initial review**
>
> Thank you for the positive review.
>
> Regarding GPU memory, as stated on page 6 "We aim to keep the training pipeline simple and all models are trained on a single Nvidia 1080Ti GPU with PyTorch." The GPU has 11 Gb memory, and handles 4 slots and a batch size of 32 when trained with $128 \times 128$ input resolution.

---

### Official Review · Reviewer_8fzT · 2023-11-07

**Soundness:** 3 good
**Presentation:** 3 good
**Contribution:** 2 fair
**Rating:** 5
**Confidence:** 4

**Summary:**

This work introduces Divided Attention, an extension of the Slot Attention Network (SAN) for unsupervised multi-object discovery in real-time. Divided Attention takes both the RGB image and optical flow as inputs, and learns a set of “slots” as encodings that can reconstruct the optical flow. By constructing the optical flow (instead of image reconstruction in typical SAN), the model can focus on separating the objects in the scene and understanding their motion, rather than overfitting to the relatively less related visual details such as illumination and texture. Another key component is an adversarially trained flow decoder, which attempts to reconstruct the entire flow from each individual slot (the main decoder reconstructs the flow only within each mask). By employing this adversarial training, the slots are encouraged to learn “contextually separated” encoding of the scene, and consequently result in separated, interpretable object representations.

**Strengths:**

- This work proposes to leverage optical flow for unsupervised multi-object discovery. In a video setting, it is more intuitive to extract information from the motion of pixels, and discover coherent regions as independently moving objects.

- Divided Attention does not require any pre-trained visual features, and thus is more flexible to be applied in various applications. Moreover, its training and inference can use a dynamic number of slots, depending on the context.

- The model is very efficient, enabling real-time inference speed.

**Weaknesses:**

- Flow input: In real-world practice, optical flow has to be obtained by running a flow estimation algorithm (e.g., RAFT). This would raise two concerns: 1) The flow estimation model is pre-trained with external knowledge and data in a supervised manner, which somehow contradicts with the claim that Divided Attention is unsupervised and requires no pre-trained features. 2) If we take the inference time of the flow estimation model into consideration, the FPS of the whole pipeline would be decreased, and achieving the real-time application would be more challenging.

- Temporal consistency: In the base version of Divided Attention, temporal consistency across frames is not guaranteed. Additional tricks (e.g., inheriting slots from previous frames, or post-processing results of multiple frames) need to be incorporated for temporal consistency. This is not desirable considering the main application is object segmentation in videos.

- Missing ablation study: It is suggested to quantitatively examine the design choices in Divided Attention via ablation study experiments. For example, $\lambda$ in the adversarial training, the model architecture, and the number of slots during training and inference, should be tested with different choices for justification and better understanding of the proposed method, Divided Attention.

**Questions:**

- In Table 1, why are the FPS of some baselines (including DyStab) not listed?

---

> ### Author Response · Authors · 2023-11-17
> **Author response to the initial review**
>
> Thank you for your thoughtful review. Here we address your concerns below:
>
> **W1:** *Flow input: In real-world practice, optical flow has to be obtained by running a flow estimation algorithm (e.g., RAFT). This would raise two concerns: 1) The flow estimation model is pre-trained with external knowledge and data in a supervised manner, which somehow contradicts with the claim that Divided Attention is unsupervised and requires no pre-trained features. 2) If we take the inference time of the flow estimation model into consideration, the FPS of the whole pipeline would be decreased, and achieving the real-time application would be more challenging.*
>
> **R1:** This is a legitimate concern, and we have addressed it in the Page 3 footnote. We adhere to the convention in unsupervised moving object segmentation, treating optical flow as an off-the-shelf module. In practice, optical flow can be determined without the need for learning [1,2].  We use RAFT [3] only for a fair comparison with baseline methods. Notably, RAFT demonstrates excellent generalization to a diverse range of visual data, and we refrain from fine-tuning the it for our target data.
>
> From an engineering perspective, as more real-time optical flow methods become accessible (including RAFT, which can also operate in real-time), DivA has the potential to be applied across a wide range of applications.
>
> [1] Determining optical flow. Horn, Berthold KP, and Brian G. Schunck. "Determining optical flow." Artificial intelligence 17.1-3 (1981): 185-203.
>
> [2] Secrets of optical flow estimation and their principles. Sun, Deqing, Stefan Roth, and Michael J. Black. "Secrets of optical flow estimation and their principles." 2010 IEEE computer society conference on computer vision and pattern recognition. IEEE, 2010.
>
> [3] Raft: Recurrent all-pairs field transforms for optical flow. Teed, Zachary, and Jia Deng. "Raft: Recurrent all-pairs field transforms for optical flow." Computer Vision–ECCV 2020: 16th European Conference, Glasgow, UK, August 23–28, 2020, Proceedings, Part II 16. Springer International Publishing, 2020.
>
> **W2:** *Temporal consistency: In the base version of Divided Attention, temporal consistency across frames is not guaranteed. Additional tricks (e.g., inheriting slots from previous frames, or post-processing results of multiple frames) need to be incorporated for temporal consistency. This is not desirable considering the main application is object segmentation in videos.*
>
> **R2:** DivA is motivated by biological vision that operates in real-time with low latency and low computational budget (see Sect. 1), so we design it to handle instantaneous motion, operating on image-flow pairs. This makes DivA suitable for online closed loop visual perception systems that receive data sequentially and run inference on-the-fly (just like biological vision). The primary goal of DivA is object discovery. In real applications like video object segmentation, once an object is discovered, one is free to apply tracking, or engage image-level feature learning.
>
> However, in applications with more computational budget and no real-time constraints, incorporating temporal consistency could enhance the system, as suggested by the reviewer. It is worth noting that this is tangential to the primary purpose of DivA, but if needed, one may also integrate DivA into such systems where it can function as a sub-module.
>
> **W3:** *Missing ablation study: It is suggested to quantitatively examine the design choices in Divided Attention via ablation study experiments. For example,  in the adversarial training, the model architecture, and the number of slots during training and inference, should be tested with different choices for justification and better understanding of the proposed method, Divided Attention.*
>
> **R3:** We do provide ablations on adversarial training in Table 2 and Figure 5, and ablations on number of slots in Table 3. Ablations on network modules, including conditional decoder and adversarial decoder, are also in Table 2 and 3.
>
> **Q1:** *In Table 1, why are the FPS of some baselines (including DyStab) not listed?*
>
> **R4:** Our emphasis is on comparing the speed of DivA with single-frame methods, aligning with the purpose of DivA. When it comes to video batch methods, the FPS (some of them not reported by the original author) may not accurately represent the running speed because of the delay introduced by collecting the batch of frames. However, we acknowledge this concern, and in the next revision, we will include information about their speed.
>
> We hope this response answers the reviewer's question. If there are further concerns, we are more than willing to provide additional explanations.

---

> > ### Comment · Reviewer_8fzT · 2023-11-22
> > **Reviewer Response**
> >
> > The reviewer appreciates the authors' helpful responses. Although the idea proposed in this work is indeed interesting for future reseach, the reviewer still has some concerns after reading the responses from the authors and comments from other reviewers:
> > - The method is proposed for handling instantaneous motion. However, it requires running an optical flow estimator first (also mentioned by Reviewer 5Uwa), and the inference efficiency comparison is rather obscure currently.
> > - The flow-based approach is only able to discover objects in motion (Reviewer i9DX). Objects, by the common understanding in vision, could be static but not considered in this work. It would be better if the problem scope is revised.
> >
> > Given this concerns, the reviewer decides to remain the current rating. Again, the authors' effort in polishing this work and providng feedback is greatly appreciated.

---

### Official Review · Reviewer_i9DX · 2023-11-07

**Soundness:** 2 fair
**Presentation:** 3 good
**Contribution:** 1 poor
**Rating:** 5
**Confidence:** 5

**Summary:**

The paper presents Divided Attention (DivA), an unsupervised method for segmenting visual fields into independently moving regions without manual supervision. The model uses an adversarial conditional encoder-decoder architecture with interpretable latent variables, building on the Slot Attention architecture. It's designed to decode optical flow using the image as context without reconstructing the image itself, thus avoiding issues with complex image variability. DivA can handle varying numbers of objects and resolutions, is invariant to object label permutations, and doesn’t require explicit regularization or pre-trained features. It achieves high run-time speed (up to 104FPS) and narrows the performance gap with supervised methods to 12% or less. The code will be made available upon publication.

**Strengths:**

+ This paper is well-written and easy to follow.
+ The model is capable of inference speeds up to 104FPS, significantly faster than current unsupervised methods.
+ The method does not rely on pre-trained image features from external datasets, enabling its use in a broader range of scenarios.

**Weaknesses:**

Firstly, I suggest the authors pay attention to the terminologies: in the title and introduction, the authors claim to do multi-object discovery/segmentation. However, they are actually doing moving object segmentation. "regions of an image whose corresponding motion is unpredictable from their surroundings" This should be the definition of moving objects but not objects. In other words, for some datasets with both moving and non-moving objects, such as MOVI-D, the proposed method will fail to work.

Secondly, I think the novelty/contribution of this work is limited. Reconstructing in the flow space with slot attention has been widely explored before, e.g. SAVI, the conditional decoder and the adversarial loss is also somewhat not quite novel.

Moreover, there is no explanation why the author wants to take flow as the input and reconstruction space but RGB as the condition rather than take RGB as the input but flow as the condition. More ablations regarding this should be conducted. Also, quantitative results for the ablation with adversarial decoder should also be reported as that's one of the claimed contributions.

Finally, more visualizations should be included, at least in the supplementary.

---------------------
Thanks again for the quick reply.

For R1, I think I understand what the authors claimed -- the definition of the ``object'' is proper and in principle, the non-moving object can be captured in real practice, with the proposed method. However, I still doubt if that's the case -- it's hard to predict the results unless seeing the results.

For R2, the discussions for those references sound promising. Though one minor thing is, for [4], the authors have ablations to verify that their method can still work without pre-trained features -- they can first train the ViT from the target dataset and then do object discovery in that space. Not to mention both [3] and [4] require no additional data (flow) but just the RGB images.

I respect the effort of the authors during the rebuttal stage and would like to slightly upgrade my rating, but will not champion this paper.

**Questions:**

See previous section.

---

> ### Author Response · Authors · 2023-11-23
> **Author response to the initial review**
>
> Thank you for your thoughtful review. Here we address your concerns below:
>
> **W1:** *Firstly, I suggest the authors pay attention to the terminologies: in the title and introduction, the authors claim to do multi-object discovery/segmentation. However, they are actually doing moving object segmentation. "regions of an image whose corresponding motion is unpredictable from their surroundings" This should be the definition of moving objects but not objects. In other words, for some datasets with both moving and non-moving objects, such as MOVI-D, the proposed method will fail to work.*
>
> **R:** There are many definitions of objects, we adopt Gibson’s definition as “layouts of surfaces surrounded by the medium” [1]. Due to parallax, such objects elicit piecewise smooth optical flow *even if they are static* just due to the viewer’s motion. Therefore, our definition is not restricted to independently moving objects, but to any scene that is non-convex and, consequently, results in piecewise smooth optical flow — with occluding boundaries — as a result of either viewer or object motion.
>
> Note that we define the object as “regions of an image” whose corresponding motion is unpredictable, we do not refer to regions of the scene. Static scenes (that do not move) result in regions of the images that move, as a result of the viewer’s motion, and so long as they are not globally convex (which no real scene is) they fit our definition.
>
> [1] James J Gibson. The ecological approach to the visual perception of pictures.
>
> **W2:** *Secondly, I think the novelty/contribution of this work is limited. Reconstructing in the flow space with slot attention has been widely explored before, e.g. SAVI, the conditional decoder and the adversarial loss is also somewhat not quite novel.*
>
> **R:** Respectfully, we argue that the “not quite novel” assessment may oversimplify the significance of our contributions. Given that slot attention was introduced three years ago, to our knowledge, it has not been widely applied to unsupervised multi-object discovery on data beyond toy examples and highly constrained data domains, whereas DivA has been rigorously tested on more generic video datasets such as DAVIS and FBMS, without restrictions on object classes or background motion. But we welcome any specific reference that the reviewer may have.
>
> Further, it is important to note that SAVI is not strictly an unsupervised method, as it requires input in the first frame, as detailed in our discussion on page 9.
>
> **W3:** *there is no explanation why the author wants to take flow as the input and reconstruction space but RGB as the condition rather than take RGB as the input but flow as the condition.*
>
> **R:** We do provide an explanation in the first two paragraphs of Section 2.2. Here we provide further insights: We believe in embodied cognition, and anything that we may want to define as an object is either something that moves independently of the surrounding (e.g., an animal) or that is at least partially surrounded by the medium (e.g., trees, houses, rocks, etc.) so that, even if static, they induce apparent motion on the image plane that our method can bootstrap into objects. If an “object” is neither moving, nor triggers occluding boundaries in response to the viewer’s motion (e.g. an object painted on a wall, as opposed to the actual object), we are comfortable not discovering it as it has no affordances such as, in Gibson’s words, graspability.
>
> As suggested by the reviewer, we conducted the experiment of taking RGB images as the input and flow as the condition, and this practice yielded unmeaningful results.
>
> **W4:** *quantitative results for the ablation with adversarial decoder should also be reported as that's one of the claimed contributions.*
>
> **R:** We do provide ablation with adversarial decoder in Table 2 and 3.
>
> **W5:** *Finally, more visualizations should be included, at least in the supplementary.*
>
> **R:** We do have figure 1, 3, 4, 6, 8, 9, 10, 11 showing qualitative visualization.

---

> ### Comment · Reviewer_i9DX · 2023-11-23
> **Response to author's rebuttal**
>
> Thanks to the authors for providing the response. However, the response still cannot resolve my concerns. I also apologize for some of the unclear comments on my original reviews.
>
> R1: I agree that the definition of objects is different in different contexts and I respect the reference of [1]. However, I don't think using this kind of definition is proper. Firstly, the definition of [1] is in terms of images/graphics rather than videos/image sequences. In the context of images, there is no definition of moving objects and static objects. Secondly, even if this definition holds, "Due to parallax, such objects elicit piecewise smooth optical flow even if they are static just due to the viewer’s motion." this is a big challenge for optical flow estimation, which is not the main focus of this paper. In other words, if the optical flow cannot distinguish those static objects, the proposed method still fails to work. Again, taking MOVI-D as an example, MOVI-D has both non-moving and moving objects with camera motion but even for the GT flow, the static objects are invisible. The proposed method will still fail to work in this dataset. I am also glad to see the results from MOVI-D if I am wrong.
>
> I think I should also rephrase my point: the terminology may be fine. Based on the definition of objects from the paper, moving objects and static objects can still be distinguished, as ideally, they both should have different optical flows compared to the background. However, distinguishing static objects from the background with, or without ego-motion is super challenging. Usually, it is very hard for an optical flow estimation model to predict the desired flow for non-moving objects, making the proposed method hard to work for discovering non-moving objects.
>
> R2: There are indeed a few works running object discovery in the real world. Here are a few works in my mind:
> [1] Simple Unsupervised Object-Centric Learning for Complex and Naturalistic Videos: https://arxiv.org/abs/2205.14065
> [2] Discovering Objects that Can Move: https://arxiv.org/abs/2203.10159
> [3] Bridging the Gap to Real-World Object-Centric Learning: https://openreview.net/forum?id=b9tUk-f_aG
> [4] Object-Centric Learning for Real-World Videos by Predicting Temporal Feature Similarities: https://openreview.net/forum?id=t1jLRFvBqm&noteId=t1jLRFvBqm
>
> R3: Thanks for the response for that. It helps a lot for better understanding.
>
> R4: I apologize for this concern. I originally meant the ablations of the $\lambda$ but I also found the results were provided in Table 2.
>
> R5: For the original comment, I meant more qualitative results for each dataset that are *similar*  to Figure 6 (several images in the same benchmark in a figure) should be included in the supplementary.

---

> ### Author Response · Authors · 2023-11-23
> **Segmenting static objects, additional related work**
>
> **R1:** *I agree that the definition of objects is different in different contexts and I respect the reference of [1]. However, I don't think using this kind of definition is proper. Firstly, the definition of [1] is in terms of images/graphics rather than videos/image sequences. In the context of images, there is no definition of moving objects and static objects. Secondly, even if this definition holds, "Due to parallax, such objects elicit piecewise smooth optical flow even if they are static just due to the viewer’s motion." this is a big challenge for optical flow estimation, which is not the main focus of this paper. In other words, if the optical flow cannot distinguish those static objects, the proposed method still fails to work. Again, taking MOVI-D as an example, MOVI-D has both non-moving and moving objects with camera motion but even for the GT flow, the static objects are invisible. The proposed method will still fail to work in this dataset. I am also glad to see the results from MOVI-D if I am wrong.*
>
> **Response:** A "detached object" is part of the scene, not of the image, regardless of whether it moves or not, thus the definition of Gibson is aligned with our use of the term. Others have also used the same, for instance [5] characterizes detachable objects based on motion discontinuities, whether the scene is static or not. It is the occlusion phenomena, a result of either object or viewer motion, that defines detached objects as manifest in images or video.
>
> As an example, see the example on a video of static scenes, taken from a moving camera (Figure 12 in the revised Appendix). We agree with the reviewer that optical flow methods sometimes fail to reliably detect occluding boundaries. This is evident in datasets like MOVI-D, where the camera angle is constrained. However, in a real embodied vision system where agents move continuously, the signal is eventually patent.
>
> **R2:** *There are indeed a few works running object discovery in the real world. Here are a few works in my mind: ...*
>
> **Response:** Thanks to the reviewer for providing the list of related work. Note that we have already referenced and cited [1] and [2] and described their limitations:
>
> [1] primarily works with synthetic data (MOVi-Solid, MOVi-E, MOVi-Tex) that has a finite set of object colors and shapes. The "real-world" data used by [1] (Youtube Traffic and Youtube Aquarium) is limited to mostly static backgrounds and specific object classes (cars and fish). On the other hand, DivA can handle more diverse video data, including unknown object appearances and classes, as well as dynamic backgrounds, as demonstrated by DAVIS and FBMS-59. This is crucial in the context of "object discovery," where the algorithm explores the unknown.
>
> [2] depends on **supervised** features [6] trained on MS-COCO to generate motion masks and trains a slot attention model on top of that. In fact, DivA can be utilized alongside [2], providing motion masks without supervision.
>
> Both [3] and [4] rely on self-supervised features (DINO) pre-trained on object-centric data. As mentioned in our manuscript,  we explore the minimal end of the supervision scale, where no purposeful viewpoint selection, 'primary object,' or human labels are used. DivA trains *ab-ovo* without any external pre-trained features, making it immune to data selection biases external to the model. This feature is also crucial for object discovery in embodied cognition. Nevertheless, we have revised the paper to include [3] and [4] in the citation as suggested.
>
> Reference:
>
> [5] A. Ayvaci et al.,  Detachable object detection: Segmentation and depth ordering from short-baseline video.
>
> [6] D. Achal et al., Towards segmenting anything that moves.

---

### Meta-Review · Area_Chair_k182 · 2023-12-12

**Metareview:**

The paper explores self-supervised object discovery in videos, by extending the established slot attention mechanism and identifying independently moving regions through optical flow. The key design is to decode optical flow using the image as context without reconstructing the image itself, thereby preventing overfitting to complex yet relatively unrelated visual details seen in prior work. An adversarial conditional encoder-decoder architecture is introduced accordingly to facilitate the learning of separated, interpretable object representations. The claimed strengths over prior work include high inference speed and the absence of a need for pre-trained image features. Empirical evaluation demonstrates the proposed method's effectiveness in handling varying numbers of objects and resolutions.

Four reviewers evaluated this paper, highlighting its major strengths as its high inference speed and independence from pre-trained image features sourced from external datasets. However, the primary concern raised by the reviewers revolves around the proposed method's capability to discover static objects. The discussion phase witnessed comprehensive exchanges between the authors and reviewers, particularly focusing on the imprecision of the notion of “objects” in the original manuscript. During this period, the authors addressed this terminology confusion by referencing Gibson's classific definition and offering a visualization (Fig. 12) illustrating parallax effects within static scenes when the camera is in motion.

The reviewers partially acknowledged the authors' clarification regarding the definition of objects, which encompasses both moving and static objects. However, skepticism persists regarding the proposed method’s ability to discover and segment static objects. This skepticism arises from the method’s reliance on off-the-shelf optical flow estimation methods, which tend to miss and fail on static objects in real-world scenarios even for their state-of-the-art version. Notably, there is a lack of corresponding empirical validation on suitable datasets, such as MOVI-D containing both non-moving and moving objects, to justify this claim. Additionally, the reviewers pointed out that the conclusion demonstrated in the specific example in Fig. 12 may not generalize, as the image shows a noticeable depth difference between the foreground object and the background in the image. Consequently, the reviewers suggested re-scoping the target problem to focus on moving object segmentation rather than object discovery. They recommended explicit discussion of potential limitations when handling objects not in motion.

Furthermore, the reviewer expressed concerns that the claimed real-time inference benefits and the handling of instantaneous motion by the proposed method may be overstated. This skepticism arises from the lack of support in the evaluation, where the run-time overhead of the optical flow estimator has not been considered in the comparison.

Additionally, the reviewers highlighted concerns about the potential incremental nature of the technical novelty in this paper. They pointed out the existence of several works that combine slot attention with the optical flow field, such as [Conditional Object-Centric Learning from Video], [Segmenting Moving Objects via an Object-Centric Layered Representation], and [Self-supervised Video Object Segmentation by Motion Grouping]. The reviewers suggested that more in-depth discussions differentiating these works from the proposed method and other related work would strengthen the paper.

During the discussion period, the authors addressed some terminology confusion and offered additional explanations. However, the primary issues outlined above persist, and these concerns are shared among the reviewers.

The ACs concur that the mismatch between the claims and empirical validation is a legitimate concern and a notable weakness in the current manuscript. Specifically, the ACs suggest either conducting a thorough evaluation to substantiate the proposed method's capability for discovering non-moving objects or reconsidering the claims made. Recognizing the validity of the weaknesses raised by the reviewers, the ACs assess that these shortcomings outweigh the current merits of the paper. Consequently, the ACs cannot recommend the acceptance of the current manuscript.

**Justification For Why Not Higher Score:**

The ACs consider that the weaknesses raised by the reviewers, particularly the discrepancy between the claims and empirical validation, are valid concerns and weigh the paper's weaknesses over the current merits.

**Justification For Why Not Lower Score:**

N/A

---

### Decision · Program_Chairs · 2024-01-16

Reject